# The Present and Future of Allergen Immunotherapy in Personalized Medicine

**DOI:** 10.3390/jpm12050774

**Published:** 2022-05-10

**Authors:** Erminia Ridolo, Cristoforo Incorvaia, Enrico Heffler, Carlo Cavaliere, Giovanni Paoletti, Giorgio Walter Canonica

**Affiliations:** 1Allergy and Clinical Immunology, Medicine and Surgery Department, University of Parma, 43121 Parma, Italy; cristoforo.incorvaia@gmail.com; 2IRCCS Humanitas Research Hospital, 20089 Milan, Italy; enrico.heffler@hunimed.eu (E.H.); giovanni.paoletti@hunimed.eu (G.P.); giorgio_walter.canonica@hunimed.eu (G.W.C.); 3Department of Biomedical Sciences, Humanitas University, 20089 Milan, Italy; 4Department of Sense Organs, Sapienza University, 00185 Rome, Italy; carlo.cavaliere@uniroma1.it

**Keywords:** personalized medicine, omics, allergen immunotherapy

## Abstract

Allergic diseases are particularly suitable for personalized medicine, because they meet the needs for therapeutic success, which include a known molecular mechanism of the disease, a diagnostic tool for that disease and a treatment that blocks this mechanism. A range of tools is available for personalized allergy diagnosis, including molecular diagnostics, treatable traits and omics (i.e., proteomics, epigenomics, metabolomics, transcriptomics and breathomics), to predict patient response to therapies, detect biomarkers and mediators and assess disease control status. Such tools enhance allergen immunotherapy. Higher diagnostic accuracy results in a significant increase (based on a greater performance achieved with personalized treatment) in efficacy, further increasing the known and unique characteristics of a treatment designed to work on allergy causes.

## 1. Introduction

The growing flood of data published in recent years on personalized medicine (PM) clearly indicates its superior capacity both in the diagnosis and treatment outcome of a wide range of pathologies, for example, in cancer, cardiovascular diseases, fertility issues and many other fields [1,2]. The term precision medicine is also frequently used (number of publications updated in March 2022, 79.37 versus 110.69 papers for personalized medicine). According to Zaim and coworkers, the development of patient-centric standards permitting to uncover clinically significant genetic abnormalities on a genome scale, is still an unaddressed challenge for the advancement of PM [3]. For an extended period, the diagnosis of IgE-mediated allergic disorders was based on in vivo tests such as skin-prick tests and on IgE antibody measurements. However, their role has diminished, since such tests often indicate sensitization and not a clinical allergy [4]. Molecular allergy diagnosis has been the first means to overcome this limit [5], and it is currently entirely accepted as a personalized diagnostic tool. This diagnostic method is based on the use of specific IgE for the identification of single allergen components (natural purified or recombinant) from multifaceted sources, such as pollen, mites, pet dander, foods and insect venoms [6,7]. A number of additional tools recently introduced, such as treatable traits and omics (which comprise proteomics, epigenomics, metabolomics, transcriptomics and breathomics), aims to predict patient response to therapies, to detect biomarkers and mediators and to assess disease control status. This review analyzes the available literature on PM for allergies, with a particular interest in allergen immunotherapy (AIT).

## 2. The Role of AIT in Allergic Diseases

Allergen-specific immunotherapy (AIT) consists of the administration, for a defined period, of escalating doses of the particular allergen against which a patient expresses IgE-mediated hypersensitivity, in order to induce a tolerance. Although subcutaneous immunotherapy (SCIT) represents the traditional and most effective route of administration, sublingual immunotherapy (SLIT) has proved to be, in recent years, a viable and safer alternative [6]. Allergic diseases are characterized by a disproportionate response to allergens that results in the IgE-mediated activation of mast cells, basophils and eosinophils with the inappropriate release of inflammatory instantaneous mediators (histamine, tryptase, chymase and proteoglycans) and the production of proteases, leukotrienes, cytokines (especially of T helper type 2) and further histamine. Contrary to conventional pharmacotherapeutical approaches (e.g., anti-histamines, anti-leukotrienes, inhaled, topical and systemic corticosteroids and biologicals), AIT does not only interfere with symptoms, it provides a long-lasting unresponsiveness to selected allergens by acting as a disease-modifying therapy [8]. The maintenance of the tolerance is accomplished through several interrelated mechanisms, including the impairment of IgE production, the switch to an increased release of IgG4, the reduction in T helper 2 cells and related cytokines (IL4, IL5 and IL13), the rise in T and B regulatory cell subsets and suppressor molecules’ expression (for instance, IL-10, IL-35, TGF-β, PD-1 and CTLA-4). All of these pathways lead to lower mast cells’ and basophils’ activation, which results in the reduction in inflammatory mediators’ release [8,9].

AIT is indicated when the correlation between allergen-specific IgE and symptoms is documented (skin-prick tests, serological measurement of IgE), when pharmacotherapy is not sufficient to control clinical manifestations and for preventing the potential onset of new sensitizations or the worsening of the respiratory disease. From a pathological point of view, AIT is indicated for the treatment of moderate-to-severe allergic rhinitis and in moderate allergic asthma. AIT for Hymenoptera venom is recommended in case of previous systemic reaction. There is not strong evidence of the effectiveness of AIT in atopic dermatitis or in food allergies, which could represent interesting arguments for future studies [9].

## 3. Treatable Traits

The diagnostic definition of a given disease is usually proposed taking into account a set of typical signs, symptoms or molecular pathways which are recurring in that group of patients but not necessarily specific to a single patient. Despite the undeniable usefulness of a classification aimed at organizing complex and heterogeneous clinical presentations, assigning labels may prove to be a simplistic approach, and it may lead to a suboptimal management of different patients belonging to the same medical category, for example, by neglecting the potential need of diverse therapeutic strategies. The “treatable traits” approach aims to detect, in a specific patient, those phenotypical or molecular traits on which it is actually possible to intervene [10]. Therefore, the concept of the “treatable traits” is meant as a new paradigm to be applied in PM in order to obtain the best management of each patient by improving the outcomes, reflecting on the research methods, generating new knowledge on the efficacy of this approach and implementing multidisciplinary models of care into practice [11].

This mindset is particularly suitable in the context of respiratory diseases, in which pathologies classified as divided (i.e., asthma and chronic obstructive pulmonary disease) may share similar endotypes and phenotypes [10]. Focusing their interest on the United Airway disease (UAD), which involves multiple phenotypes and endotypes of asthma and allergic rhinitis, sinonasal diseases and lower airway diseases, Yii et al. suggested a treatable-trait approach to its classification and management [12]. The UAD treatable traits were analyzed concerning a context including airway inflammation, impaired airway mucosal defense and exogenous cofactors (allergic sensitizers, tobacco smoke, microbes). The authors concluded that the evaluation of treatable traits is needed to appreciate the precise treatments and accomplish better outcomes in UAD patients. No studies comparing the outcome of AIT with usual methods to that based on the treatable traits of AIT are available yet.

## 4. Omics

To establish the definition of genomics, as the complete nucleotide sequence of an organism, a large number of other omics has been proposed, such as proteomics (the complete proteins of a cell in any organism), epi-genomics (the modification of nucleotides in an organism), metabolomics (the changes in gene activity in response to metabolites), breathomics (the multidimensional molecular analysis of exhaled breath) and transcriptomics (the genes’ capacity to generate different transcripts through alternative splicing). Studies investigating the optimization of diagnostic and therapeutic standards of allergic diseases by means of biomarkers developed through omics technologies were recently published. A review by Breiteneder et al. found that some of them achieved a better classification of distinct phenotypes or endotypes and increased the necessity to use biomarkers for patient selection, prediction of results and monitoring [13]. Ogulur et al. [14] stated that such biomarkers are valuable parameters as they make information on the disease endotypes, clusters, identification of treatment targets and monitoring of efficacy available. These powerful omics technologies, together with integrated data analysis, are useful in identifying clinical biomarkers; these, however, need to be precisely measured by solid and reproducible methods. The search for novel biomarkers of allergic diseases resulted in promising biomarkers of type-2 allergic diseases, including sputum eosinophils, exhaled nitric oxide and serum periostin. Biomarkers such as pro-inflammatory mediators, eicosanoid molecules, epithelial barrier integrity and microbiota changes quantified in serum, exhaled air and body fluids are suitable for diagnosing and monitoring allergies and, particularly, the efficacy of AIT, also considering the perspective of the COVID-19 pandemic [14].

### 4.1. Proteomics

Since the early diagnosis and prognosis of allergic rhinitis is concerned with accuracy problems, proteomics technology was tested by Pu X et al. Such an approach resulted in a quick, sensitive and high-throughput technology platform for early detection, therapeutic targeting and disease prognosis [15].

### 4.2. Epigenomics

A covariate-adjusted epigenome-wide association meta-analysis including pathway and regional analyses of results detected 700 DNA methylation sites, which were related to 505 genes significantly cross-sectionally associated with atopic sensitization in children, to 905 genes for environmental subjects and to 36 genes for food-allergen sensitization. Different methylations across multiple genes were found for the three phenotypes, including genes concerning innate immunity, eosinophilic esophagitis, sinusitis, atopic march and asthma. Furthermore, most of the associated methylations include all three phenotypes [16]. Further studies investigating the role of epigenomics in allergy immunotherapy are available. Based on the advances in sequencing technologies allowing to increase the information on epigenetic modifications in T cells and epigenome maps, combined with mechanistic studies, a substantial effect on phenotypic stability and function of lymphocytes was found, demonstrating that T cells undergo extensive epigenome remodeling. Moreover, the authors focused on DNA methylation, histone modifications and chromatin structure as the central epigenetic mechanisms involved in controlling T-cell responses, discussing the effect on imprinting T-cell epigenomes and the possible consequences for immunotherapy [17,18]. Furthermore, a pilot study on patients allergic to grass pollen and house dust mites treated with dual sublingual immunotherapy suggested that such treatment could be effective and that long-term tolerance to the allergens administered by AIT could be induced by epigenetic modifications of Foxp3 in memory regulatory T cells [19]. Another recent investigation found that in food allergy, epigenetic mechanisms, as well as reducing genetic modifications, can also produce modifications in immune genes, influencing, for example, immunomodulation and possibly explaining the sustained responsiveness or unresponsiveness during immunotherapy triggered by epigenetic modifications in key genes that induce tolerance of a number of foods [20].

### 4.3. Metabolomics

Since a link has been recognized between the metabolic grade of T cells and macrophages, and chronic inflammation phenotypes such as allergic inflammation, a better understanding of the pathways of these immune cells could help to reinstate and modulate their functions. This peculiar relationship can be studied by metabolomics, transcriptomics and proteomics [21].

Studies on the role of metabolomics showed that, in serum samples collected from 29 healthy controls and 72 patients allergic to dust mites, including 30 mild and 42 moderate-to-severe allergic rhinitis (AR) patients, metabolomics allowed the researchers to distinguish the different categories, suggesting that its profiling may offer novel understandings of the pathophysiological mechanisms of dust mite allergy [22]. In another study by Yuan et al., the combination of microbiome and metabolomics analyses identified important candidate biomarkers in patients with AR as differential genera of microbes and differential metabolites that could be potentially used as biomarkers for the diagnosis of AR [23]. A review of 23 studies assessing asthma or wheezing and 6 studies assessing allergy endpoints, altered metabolic pathways revealed some of the underlying biochemical mechanisms triggering these common childhood disorders, which have potential value in clinical practice, reinforcing the evidence from metabolomics studies of childhood atopic diseases [24]. Three studies addressed AIT; the first was a randomized, placebo-controlled trial administering a sublingual *Phleum pratense* extract for two years to 47 patients allergic to grass-pollen, in which metabolomics and transcriptomics were also analyzed; in the 31 patients completing the trial, the differences in the patients’ sensitization profile were associated with differential omics profiles, with better outcomes in monosensitized than in polysensitized patients after two years of treatment; a transcriptomic signature associated with effector cell downregulation suggested that SLIT has a substantial effect on crucial cellular mechanisms [25]. The second study evaluated the outcome, in AR patients with dual sensitivity, to *Dermatophagoides pteronyssinus* and *Dermatophagoides farinae* treated with SCIT, adding a metabolomics assessment to standard clinical parameters. Both treatments had therapeutic effects with no significant differences in efficacy, while a reduction in inflammation-related metabolites was constantly observed in patients undergoing SCIT treatment, highlighting arachidonic acid and its metabolites as potential biomarkers [26]. In the latter study, 68 patients completing the SLIT course were categorized into effective and ineffective groups. A total of 539 metabolites was obtained, 197 of which were recognized as known substances, whose scrutiny by Orthogonal Partial Least Square Discriminate Analysis (OPLS-DA) was used to assess the metabolite differences between the two groups. The results showed distinctive metabolite signatures and metabolic pathways, suggesting that in AR patients, the discriminative metabolites and connected metabolic pathways contribute to an improved understanding of SLIT mechanisms [27]. A further object of study was the analysis of nasal secretions from asthmatic patients to detect secretoglobin1A1 (SCGB1A1) and IL-24 protein levels during a three-year course of AIT, followed by RNA extraction to be subjected to whole transcriptome analysis. AIT inhibited pro-inflammatory response by CCL26mRNA expression, while SCGB1A1, IL7, CCL5, CCL23 and WNT5BmRNAs were induced independently of the asthma status and allergen season. This suggests that a so-far unidentified local gene expression footprint in the lower airways reveals SCGB1A1 as a novel anti-inflammatory mediator of long-term AIT [28].

### 4.4. Transcriptomics

Advances in the single-cell RNA sequencing (scRNA-seq) technique made it possible to observe the transcriptomes of single cells in patients with allergic inflammation. Using a recently published scRNA-seq study of tissue T cells as an example, it was possible to provide directions for future research and elucidate the T-cell heterogeneity occurring in an allergic inflammatory tissue focused on eosinophilic esophagitis, a prototype making possible to probe the pathogenesis of allergic inflammation at the tissue level through endoscopic biopsy specimens [29]. The metabolomics analysis detected numerous serum biomarkers able to accurately predict the efficacy of sublingual immunotherapy (SLIT) in patients with allergic rhinitis [27].

A transcriptomics assessment in pollen-allergic patients during peak season compared with untreated patients revealed that AIT induced a local gene-expression footprint in the lower airways hitherto unknown in actively traded patients. The authors proposed that this may have resulted from multiple regulatory pathways and/or Secretoglobin1A1 as a new anti-inflammatory mediator of long-term AIT in the local environment [28]. Starchenka et al. performed a transcriptome analysis and safety profile of the early response to AIT with a grass allergoid. The results showed that a higher cumulative allergoid regime was well-tolerated and safe, and that molecular markers (IL-27, IL-10, IL-4, TNF, IFNγ, TGFβ and TLR4) were the key predicted molecular drivers of the gene expression changes succeeding the treatment [30].

### 4.5. Breathomics

Breathomics have been extensively studied, particularly concerning asthma. Their role was summarized by a systematic review including twenty studies, which almost unanimously reported their capability to distinguish samples from healthy controls from those with asthma and to phenotype the disease with moderate to high accuracy. Still, the concordance in the complexes upon which discriminatory models were based was insufficient. The authors concluded that successful validation of breathomics is needed before it can be considered a standard tool for PM [31]. One year later, the ALLIANCE cohort study confirmed the negative results, stating, “Despite recent publications, we are not close to finding a clinically valuable breath volatile organic compounds biomarker for asthma or asthma phenotypes” [32].

## 5. Present Landscape of Personalized AIT

Allergic diseases can be successfully treated with many symptomatic drugs, but only the AIT acts on the causes of allergy and modifies their natural history [33,34]. A critical point is represented by the selection of the type of extract in AIT, because it induces a tolerance against a precise allergen only. In this context, Component Resolved Diagnosis (CRD) allows one to determine, through a singleplex (the sample is used to detect a single allergen) or a multiplex assessment (a sample to detect more than 100 allergens contemporaneously), whether the allergic disease is caused by a reaction to a genuine allergen or by a cross-reaction [8]. In fact, a negative factor that influenced the outcome of AIT above all in the past was the poor quality of the allergenic extracts, which was then overcome by the strict rules of the regulatory agencies to authorize the registration of AIT products, needing protein-content measurement, whole allergenic activity and major allergen content in combination with the improvement of the manufacturing level [35].

By focusing the interest on the most recent studies, thus not affected by insufficient quality, Liu et al. found that the factors associated with unsatisfactory response to subcutaneous AIT in a Chinese population were represented by coexisting atopic dermatitis, polysensitization, allergies to cats, *Alternaria* or microphytes and extended duration of allergic disease [36]. Gao et al. suggested that, in patients undergoing sublingual immunotherapy for dust mites, the inadequate allergen dosage is apparently the primary cause of treatment failure with six months of duration, being time a critical point for efficacy assessment and for a possible dose adjustment that could result in enhanced efficacy [37].

Today, the ability of PM in the diagnosis of allergic diseases and in tailoring AIT is recognized as significantly superior to that of the traditional method; however, as reported above, the AIT studies carried out so far are limited in number. Therefore, it is not possible to draw consistent conclusions [38]. Furthermore, the not-uncommon finding of non-uniform data regarding the study’s inclusion criteria, the type of AIT (subcutaneous or sublingual), the type of allergen extract administered and the treatment protocol can make it challenging to interpret the results. A further means of analyzing the two approaches would be the face-to-face comparison between conventional AIT and PM-based AIT, which, if it fully confirms the latter’s superiority, is likely to lead prescribers towards an increasingly widespread use of personalized AIT.

## 6. Future Landscape of Personalized AIT

The relevance of investigating distinctive biomarkers (meant as molecular mechanisms for diagnostic tools, prediction of response and follow up) and recognizing treatable traits appears to be clear in performing targeted AIT. As regards potential serological biomarkers, it is proved that immunotherapy induces IgA (especially in SLIT) and different subclasses of IgG (especially in SCIT) and, in particular, an IgG2 response playing a role in the competition with IgE specific for a given allergen [39]. In a recent study by Bordas-Le Floche et al., a correlation between the clinical benefits of immunotherapy in patients treated for house dust mite and their serological levels of specific IgG2, IgE and IgG4 was found [40]. Taking into account the biotechnological developments, it is also remarkable the fundamental role that omics will surely play in the characterization of more precise biomarkers for AIT, helping to correct, in future trials, the bias concerning non-uniform data and treatment protocols.

Moreover, new routes of administration are currently being researched, which could perform a new impact on meeting patients’ needs and prerogatives, improving the adherence to therapy; however, they still require more detailed studies. Intralimphatic immunotherapy (ILIT), for example, may result in a reduction in the frequency of administration; epicutaneous immunotherapy (EPIT) may open new scenarios for the treatment of food allergy especially in children [41], and the anti-inflammatory effect of local nasal immunotherapy (LNIT) with FIP-fve peptide and denatured mite *Tyrophagus putrescentiae* may be exploited for the treatment of airway allergic diseases [42,43] (Figure 1).

Furthermore, the experience and knowledge acquired in AIT may represent, in the future, a model for many other medical branches, such as oncology.

## Figures and Tables

**Figure 1 jpm-12-00774-f001:**
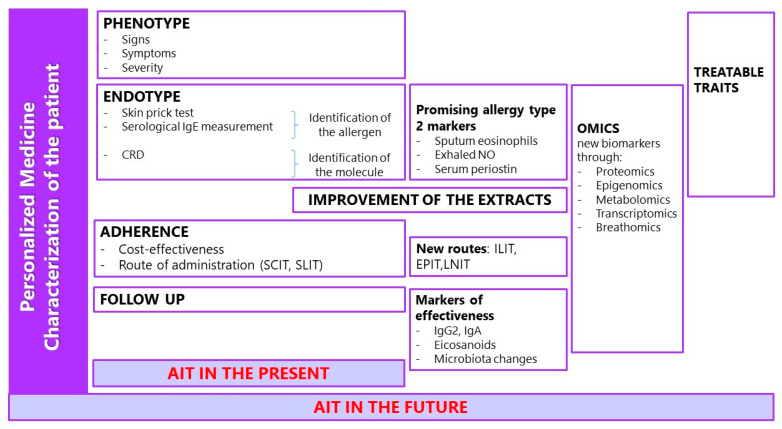
Present and future perspectives of personalized medicine in allergen immunotherapy (AIT). Abbreviations: CRD, Component Resolved Diagnosis; NO, nitric oxide; SCIT, subcutaneous immunotherapy; SLIT, sublingual immunotherapy; ILIT, intralimphatic immunotherapy; EPIT, epicutaneous immunotherapy; LNIT, local nasal immunotherapy.

## Data Availability

Not applicable.

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
