# Peer review of "The Present and Future of Allergen Immunotherapy in Personalized Medicine"

_jpm, 2022, doi:10.3390/jpm12050774_

Round 1

Reviewer 1 Report

The mini-review “The Present and Future of Allergen Immunotherapy in Personalized Medicine” of Incorvaia, C. et al. focuses on current problems of personalized allergen-specific therapy, the biomarkers, and mediators as predictors of patient’s response to the potential therapeutic approach. Molecular allergy diagnostic is an important tool for personalized AIT, the authors discussed in detail in a previously published review Incorvaia C, Al-Ahmad M, Ansotegui IJ, et al. Personalized medicine for allergy treatment: Allergen immunotherapy still a unique and unmatched model. Allergy. 2021;76:1041–1052. https://doi.org/10.1111/all.14575. Here the authors also discuss other current tools such as omics (proteomics, epigenomics, metabolomics, transcriptomics, and breathomics) and initiate the further targeted investigations for the global use of personalized AIT. Moreover, similar approaches can be used for monitoring prescribed therapy.

Author Response

The manuscript has been completely reworked and revised with a special focus on the English language. 

Reviewer 2 Report

Incorvaia et al review the literature on personalized medicine with a focus on allergy and AIT. The review is well written, well referenced, and is of interest for both the PM community as well as the AIT community and I thus support the concept of this review in general. However, the review could benefit from major revision.

  • This review has some overlap with the following publication: Incorvaia, C. et al,. Allergy 2021, 76, 1041-1052, doi:10.1111/all.14575. As the authors here intended to focus more on PM, it would be good to discuss the specific standing of AIT in PM in greater detail. How does AIT relate to PM compared to other diseases/therapies? How advanced is PM in AIT compared to other diseases/therapies? What are the challenges to overcome for PM in AIT and other diseases/therapies?
  • The information in this review is very disjointed which does not invite the reader into the topic. Major re-structuring is suggested, more segments and a more clear setup. If possible, the information should be presented in a more concise way.
  • Even though the focus is on PM, it would be good to have a short introduction on allergy/AIT including the most important indications and hallmarks of successful therapy, and the current challenges of AIT. For example, regulatory T/ B cells, total IgE versus specific IgE, IgG4, and basophils are not introduced or discussed.
  • The “Omics” part could be broken down into different sections for example based on omics method, or subtype of disease, or type of AIT, type of biomarker... Each subsection could then by quickly summarized to help guide the reader through.
  • At the end, if possible, it would be good to summerize/formulate some key concepts or take home messages, which could also be displayed in a figure/scheme/table. 
  • Since the field is fast evolving, the “future landscape” could be more detailed and could be considering potential novel PM methods, novel promising AIT biomarkers or mechanisms that could be applied for PM. Furthermore it could be discussed in more depth how the current challenges of AIT could be advanced by PM in the future and how AIT could advance PM in other diseases/therapies.

- Check Reference 11

Author Response

The paper has been completely revised and rearranged in order to bring the text more into line with the suggestions of the reviewer. In particular more segments have been included, such as the introduction on allergy/AIT and their immune mechanisms. "Omics" has been divided in subsections. "Landscape" section has been divided in two more detailed subsections  (present and future), improving the part concerning biomarkers and routes of administration. Finally, key concepts have been displayed in a figure. Moreover, the text has been completely revised for what concerns English language. 

Round 2

Reviewer 2 Report

The authors have revised their review, added detail and structure to the manuscript and included a figure that summarizes the main points. I support the publication of this article in its current form. Some minor suggestions for further improvement:

- A final, detailed spell- and grammar check is suggested as some mistakes have come up in the revised document. Examples: cathegory (category). It is proved that immunotherapy induces IgA (It is proven that/it was shown that)

- Some sentences could be shortened for improving clarity. For example, lines 1-8 of "treatable traits" is a long and complex sentence that could be shortened without losing much information. 

- Avoiding duplication of keywords and title could improve visibility